# *Plantago major* and *Plantago lanceolata* Exhibit Antioxidant and *Borrelia burgdorferi* Inhibiting Activities

**DOI:** 10.3390/ijms25137112

**Published:** 2024-06-28

**Authors:** Pille-Riin Laanet, Olga Bragina, Piia Jõul, Merike Vaher

**Affiliations:** Department of Chemistry and Biotechnology, Tallinn University of Technology, Akadeemia tee 15, 12618 Tallinn, Estonia; pille-riin.laanet@taltech.ee (P.-R.L.); olga.bragina@taltech.ee (O.B.); piia.joul@taltech.ee (P.J.)

**Keywords:** Lyme disease, *Borrelia burgdorferi*, phytochemicals, antioxidants, antibacterials, biofilm inhibition

## Abstract

Lyme disease, caused by *Borrelia burgdorferi sensu lato* infection, is the most widespread vector-borne illness in the Northern Hemisphere. Unfortunately, using targeted antibiotic therapy is often an ineffective cure. The antibiotic resistance and recurring symptoms of Lyme disease are associated with the formation of biofilm-like aggregates of *B. burgdorferi*. Plant extracts could provide an effective alternative solution as many of them exhibit antibacterial or biofilm inhibiting activities. This study demonstrates the therapeutic potential of *Plantago major* and *Plantago lanceolata* as *B. burgdorferi* inhibitors. Hydroalcoholic extracts from three different samples of each plant were first characterised based on their total concentrations of polyphenolics, flavonoids, iridoids, and antioxidant capacity. Both plants contained substantial amounts of named phytochemicals and showed considerable antioxidant properties. The major non-volatile constituents were then quantified using HPLC-DAD-MS analyses, and volatile constituents were quantified using HS-SPME-GC-MS. The most prevalent non-volatiles were found to be plantamajoside and acteoside, and the most prevalent volatiles were β-caryophyllene, D-limonene, and α-caryophyllene. The *B. burgdorferi* inhibiting activity of the extracts was tested on stationary-phase *B. burgdorferi* culture and its biofilm fraction. All extracts showed antibacterial activity, with the most effective lowering the residual bacterial viability down to 15%. Moreover, the extracts prepared from the leaves of each plant additionally demonstrated biofilm inhibiting properties, reducing its formation by 30%.

## 1. Introduction

### 1.1. Lyme Disease

Ticks inhabiting temperate geographic regions can carry a variety of pathogens; among them, a group of bacteria named *Borrelia burgdorferi sensu lato*. Currently, the *sensu lato* complex comprises at least twenty proposed or confirmed species worldwide, nine of which have been found to infect humans [1]. Out of these, four species, namely, *B. burgdorferi sensu stricto*, *B. afzelii*, *B. garinii*, and *B. bavariensis*, cause a great majority of human Lyme disease cases [1]. Lyme disease, or borreliosis, is a major public health concern. Already the most common vector-borne illness in the Northern Hemisphere, the infection rates continue to increase globally [2]. The clinical manifestations of Lyme disease are diverse. The most common symptom of *B. burgdorferi* infection is an erythema migrans rash at the site of the tick bite [3]. Other symptoms include headache, mild stiff neck, general fatigue, fever, arthralgia, myalgia, and lymphadenopathy [4]. While most Lyme disease cases are successfully cured with antibiotics, 10–20% of patients report lingering symptoms like fatigue, cognitive impairment, joint and muscle aches, and/or depression, referred to as post-treatment Lyme disease [5]. The pathogenesis of chronic Lyme disease is still being debated, with the discussion focusing on the concepts of persistent infection and/or autoimmunity [6]. According to previous research, chronic illness is thought to be tied to the development of resistant bacterial forms [6,7]. *B. burgdorferi* are pleomorphic bacteria that can change their morphology as a response to inhospitable environmental conditions, like the administration of antibiotics [2]. In addition to the typical spirochaetal forms, *B. burgdorferi* can also form spherical shapes, blebs, detaching granules or pearls, and agglomerations of spirochaetes into biofilm-like colonies [8,9,10,11]. There is plausible evidence that pleomorphism may help to evade the immune system or decrease susceptibility to antibiotics [7,12]. Therefore, it is suggested that novel treatment approaches should consider the specific efficacy as related to the persistent morphological forms of the bacteria.

### 1.2. Ethnomedicinal Use and Therapeutic Properties of the Plantago Species

Ethnomedicine has long recognised plant extracts as a valuable source of antioxidant and antibacterial agents. These useful therapeutic properties are exhibited by a variety of phytochemicals, the bioactive secondary metabolites [13]. Research conducted by our group and others suggests that some plant extracts could also exert promising effects in the fight against Lyme disease, specifically inhibiting *B. burgdorferi* and its latent forms [14]. As a part of our research into native Estonian plants with known antioxidative and/or antibacterial properties to evaluate their potential as inhibitors of *B. burgdorferi*, two species from the *Plantago* genus, *Plantago major* and *Plantago lanceolata*, were investigated. *Plantago* species are widely considered to be weeds, yet they have been used as medicinal plants for centuries [15]. Both *P. major* and *P. lanceolata* have a history of a variety of ethnomedicinal applications due to their diverse properties. *P. major* is one of the most common medicinal herbs in the world, and its leaves have been widely used to promote wound healing. Additionally, *P. major* is used to treat skin infections and other infectious diseases, as well as digestive and respiratory disorders; to enhance the circulation and reproduction; to relieve pain and fever; and to prevent cancer [16]. The aerial parts of *P. lanceolata* are thought to possess wound healing, anti-inflammatory, antibacterial, diuretic, and anti-asthmatic effects [17]. *P. lanceolata* leaves have been used to treat arthritis, as well as mouth, throat, and upper respiratory tract conditions. They have also been used topically to treat skin diseases [18,19]. 

The research into discovering and confirming the full therapeutic potential of *Plantago* extracts remains partially inconclusive. There have been a number of scientific efforts to evaluate the antioxidant, anti-inflammatory, and wound-healing activities of *Plantago* extracts and isolated compounds. Previous studies have confirmed that *Plantago* species are rich in natural antioxidants, and their methanolic and ethanolic extracts exhibit reactive oxygen species (ROS) scavenging activity [20]. This property demonstrates their potential as agents against many chronic diseases, including age-related cardiovascular and neurological diseases as such illnesses are influenced or brought on by oxidative stress, or in other words, by the abundance of ROS and other free radicals in the body [21]. However, the results demonstrating the antioxidant potential of *Plantago* species are difficult to compare due to the different extraction protocols and analysis methods used [20]. Several studies have been conducted to confirm the anti-inflammatory properties of *Plantago* species, particularly *P. lanceolata* [20]. As inflammation is a pathological component of many autoimmune diseases, as well as cancer, and as it might play an important role in Lyme disease pathogenesis, research into effective anti-inflammatory drugs is an important field in drug discovery [22]. Both *P. major* and *P. lanceolata* extracts have been demonstrated to exhibit anti-inflammatory and immunoenhancing properties [22,23,24,25]. Additionally, the wound-healing properties of extracts from both *P. major* and *P. lanceolata* leaves have been scientifically proven by several authors [19,26,27,28,29]. Thus, there is already evidence of several therapeutically important qualities of *P. major* and *P. lanceolata* extracts. However, the antibacterial properties of *Plantago* species are not as thoroughly researched.

### 1.3. Phytochemical Screening of Plantago Extracts

The main analytical methods that have been used to detect and quantify phytochemicals in *Plantago* extracts have been HPLC, CE, GC, and MS. RP-HPLC has been used to measure aucubin content, flavonoids, and phenolic acids [30,31]. UPLC-MS/MS has been used to compare the phenylethanoid glycoside profiles [32]. CE-MEKC has been used to quantify two iridoid glycosides, namely, aucubin and catalpol, and two phenylethanoid glycosides, namely, acteoside (or verbascoside) and plantamajoside [33]. LC-MS/MS in the MRM mode was used to measure plantamajoside and acteoside [34]. Therefore, it has been established that HPLC-MS/MS is a suitable analytical method for the detection of non-volatile phytochemicals of *Plantago* extracts. GC-MS has previously been used to identify the volatiles in *Plantago* extracts. GC-MS was utilised to analyse the content of triterpenes, lipids, and amino acids in the methanolic extract of *P. major* leaves [35] and to identify the various phytochemicals extracted from both *P. major* and *P. lanceolata* leaves and fractioned using different solvents [36]. Thus far, the determination of volatiles in the two *Plantago* species has not been well covered in previous research. As demonstrated by our group, headspace solid-phase microextraction (HS-SPME) coupled with GC-MS is well suited to determine and compare the composition of volatile compounds in plant materials [37]; it was therefore also employed in this study.

Phytochemical analyses have revealed that the main bioactive compounds present in the aerial parts of the *Plantago* species are phenylpropanoid glycosides, iridoids, triterpenes, flavonoids, and phenolic acids [15]. Phenylethanoid glycosides, most prevalently acteoside and plantamajoside, are the key metabolites in the *Plantago* species [15]. Rønsted et al. have reviewed the iridoid glycosides and caffeoyl phenylethanoid glycosides detected in *Plantago* species. According to their research, *P. major* has been found to contain aucubin, melittoside, asperuloside, melampyroside, plantarenaloside, ixoroside, majoroside, 10-hydroxymajoroside, 10-acetoxymajoroside, geniposidic acid, gardoside, acteoside, and plantamajoside [20]. Rønsted et al. also reported that *P. lanceolata* has been found to contain aucubin, catalpol, acteoside, asperuloside, globularin, methyl deacetulasperulosidic acid, isoverbascoside, plantamajoside, and lavandufolioside [20]. Turgumbayeva et al. studied the volatiles in the methanolic extract of *P. major* leaves and found mainly lupeol, benzimidazo[2,1-a]isoquinoline, palmitic acid, and β-amyrin [35].

### 1.4. Specific Antibacterial Potential of Plantago Extracts

Rahamouz-Haghighi et al. fractioned both *P. major* and *P. lanceolata* leaf extracts using different solvents [36]. Using GC-MS, they identified the volatile phytochemicals in these extracts, many of which are known for their antibacterial properties. Additionally, the same paper demonstrated that the leaf extracts of both *P. major* and *P. lanceolata* showed varied inhibitory effects against Gram-positive and Gram-negative bacterial strains. As natural antioxidants, especially polyphenols, are often found to be valuable antibacterial agents [38], it is likely that the non-volatiles found in *P. major* and *P. lanceolata* could also exhibit antibacterial properties. Thus far, there have been no studies evaluating the effectiveness of *Plantago* extracts in inhibiting *B. burgdorferi*. When it comes to the fight against Lyme disease, it would also be important to determine if the antibacterial effects are similarly evident against *B. burgdorferi* biofilm. It is already known that phytochemicals can be valuable biofilm inhibiting agents, operating through different mechanisms to antibiotics [39]. However, the potential of *Plantago* species in this field has not yet been explored. As a final consideration, it would be very beneficial to be able to obtain a natural agent inhibiting *B. burgdorferi* from the same geographic region where Lyme disease is presenting as an increasingly pervasive public health concern. Considering these unexplored hypotheses, this paper demonstrates the potential of *P. major* and *P. lanceolata* extracts to inhibit *B. burgdorferi* and its biofilm and thus offers a potential novel therapeutic approach in the fight against Lyme disease.

## 2. Results and Discussion

### 2.1. Antioxidative Activity and Colorimetric Analyses

The plant samples analysed in this study and the respective abbreviated names used in this paper are given in Table 1. The abbreviated names were given based on the plant species (PM for *P. major* and PL for *P. lanceolata*) and the part of the plant utilised (F for leaves and H for the whole herb), and the final letter represents the place of origin of the plant material.

The antioxidant activity and total polyphenolic, flavonoid, and iridoid content of each of the six *Plantago* samples are given in Table 2. All *Plantago* samples showed strong antioxidant capabilities, up to 1.83 ± 0.32 mmol TE/g for PLFM. The next highest results were those of the extracts of PMFM, PLFJ, and PMHK, with antioxidant activities of 1.51 ± 0.04, 1.48 ± 0.34, and 1.20 ± 0.26 mmol TE/g, respectively. In the investigation of the antioxidant activities of 30 different plant extracts of industrial interest published by Dudonné et al., the antioxidant capacities of the aqueous extracts ranged from 0.18 to 8.52 mmol TE/g [40]. Although there were a few samples with antioxidant capacities higher than 4 mmol TE/g, most of the plants studied showed antioxidant properties in the range of 0.2 to 1.6 mmol TE/g [40]. Thus, as the antioxidant capacities of the *Plantago* samples in this study ranged from 0.6 to 1.8 mmol TE/g, there was considerable antioxidant potential detected. Dalar et al. have previously found that Turkish *P. lanceolata* leaves exhibited antioxidant content of 1.6 mmol TE/g, and the whole plant exhibited antioxidant content of 0.9 mmol TE/g, which are comparable to our results [41]. Dalar et al. also reported that “the ORAC values of *P. lanceolata* leaf were higher than those of rosehip (1085 ± 24.32 μmol TE/g), cinnamon (1069 ± 5.47 μmol TE/g), oregano (1233 ± 41.36 μmol TE/g) and nutmeg (1187 ± 8.74 μmol TE/g) – herbs known for their high antioxidant capacities and health benefits” [41].

The polyphenolic content in the extracts prepared from self-gathered plant materials was considerably higher than that of those made from purchased plant materials (PMFK, PMHK, PLHS), with PLFM containing the highest number of polyphenolics (55.5 ± 4.8 mg GAE/g), followed by PLFJ and PMFM (45.9 ± 0.8 and 32.7 ± 2.4 mg GAE/g, respectively). The total flavonoid content followed a similar trend as the total polyphenolic content, with PLFJ (14.0 ± 0.7 mg QE/g) and PLFM (12.4 ± 1.2 mg QE/g) showing the highest concentrations, followed by PMFM (8.3 ± 0.2 mg QE/g). Additionally, the colorimetric assay confirmed that sample PLFM contained the highest concentration of iridoids, at 23.4 ± 4.6 mg AE/g. For other extracts, the iridoid content was considerably lower. The next highest concentration was found in the extract of PLHS (14.3 ± 1.4 mg AE/g), and the one after that in PMHK (11.4 ± 0.9 mg AE/g). Our results confirm the earlier reports of *Plantago* species containing a significant volume of polyphenolics and iridoids and exerting considerable antioxidant activity [15,41]. Additionally, there was a correlation noted between the polyphenolic content and the antioxidant activity of the extracts (R^2^ of the linear regression 0.797), which is explained by the recognised antioxidant capacity of polyphenolic phytochemicals [40,42]. Overall, when considering the mean values for each plant in the results of the different colorimetric tests, the bioactive content seems to be higher in *P. lanceolata* extracts than in *P. major* extracts.

### 2.2. Identification and Quantification of Non-Volatile Phytochemicals

The chromatograms of all six *Plantago* extracts are presented in Figure 1. All quantified compounds were chosen based on the respective relative peak areas (RPAs) in the HPLC-DAD experiment given that at least one of the extracts showed a peak with an RPA equal to or higher than 0.2 at that timepoint. Based on this selection, there were a total of 40 unique constituents quantified. In addition to the retention time on the chromatogram, the MS/MS fragmentation spectra were considered when matching up phytoconstituents of a similar retention time from different extracts. Only the two most significant peaks on all chromatograms—plantamajoside (Rt = 10.9 min) and acteoside (Rt = 11.5 min)—were identified and quantified using reference standards (Figure 1). All other peaks were quantified using one of reference standards available, matching the peak’s absorption spectra with the most similar example from the standard compounds. The reference standards, along with their respective retention times and regression curves, were the following: chlorogenic acid (Rt = 8.2 min, y = 0.0129x + 0.0227); plantamajoside (Rt = 10.9 min, y = 0.0062x + 0.016); acteoside (Rt = 11.5 min, y = 0.0073x + 0.0433); luteolin (Rt = 16.4 min, y = 0.0424x + 0.0177); apigenin (Rt = 18.3 min, y = 0.0355x + 0.0571). All quantified peaks, the respective absorption and quantification data, and the corresponding molecular ion and MS/MS fragmentation patterns are given in Table 3.

The highest combined concentration of phytochemicals was found to be in the extract of PLFM, amounting to 4694.1 ± 211.3 mg/L (corresponding to 93.9 ± 4.2 mg/g in the air-dried plant material), followed by another extract of *P. lanceolata* leaves, PLFJ, at a total concentration of 4074.8 ± 150.3 mg/L (81.5 ± 3.0 mg/g), and then by the extract of *P. major* leaves, PMFM, at 2934.0 ± 103.4 mg/L (58.7 ± 2.1 mg/g). PMHK (1572.9 ± 57.9 mg/L, 31.5 ± 1.2 mg/g), PLHS (1268.5 ± 83.1 mg/L, 25.4 ± 1.7 mg/g), and PMFK (1100.0 ± 42.7 mg/L, 22.0 ± 0.9 mg/g) extracts showed lower concentrations of non-volatile phytochemicals. Thus, the total phytochemical content shows a distinction between the freshly gathered and the purchased plant material, as the latter contained significantly lower quantities of phytoconstituents. Amongst all quantified compounds, plantamajoside was the one found in the highest concentration amongst *P. major* extracts, with the highest concentration being 1693.0 ± 65.6 mg/L (33.9 ± 1.3 mg/g) in the *P. major* leaves extract, PMFM. Acteoside was the compound found in the highest concentration amongst *P. lanceolata* extracts, with the highest concentration being 3309.0 ± 249.8 mg/L (66.2 ± 5.0 mg/g) in the *P. lanceolata* leaves extract, PLFM, followed by 1652.6 ± 163.5 mg/L (33.1 ± 3.3 mg/g) in the *P. lanceolata* herb extract, PLFJ. When comparing the chromatograms and the phytochemical content within the extracts of the same species, there are some clear differences in both the chromatographic profile and the concentrations of the constituents. However, the main distinction would be the relative difference in the amount and content of phytochemicals between the self-gathered and purchased material. Regarding the variety of quantified compounds, the *P. lanceolata* extract PLFJ stands above the rest at 23 unique phytochemicals, followed by PMFM and PLFM, both at 13 each, and then by PMHK at 8, PLHS at 7, and PMFK at 6. Therefore, both the total quantity and the variety of non-volatiles in the extracts are considerably different based on the method of acquisition of the raw material.

The two phytochemicals found in the highest concentrations, plantamajoside and acteoside, have both been demonstrated to exhibit beneficial properties. Plantamajoside has been reported to exhibit antibacterial, antiallergic, anti-inflammatory, antioxidant, and enzyme inhibitory activities [42]. Acteoside has been shown to exert antioxidant, anti-inflammatory, antimicrobial, and anticancer effects [43]. Although any specific effects demonstrated in our work by the whole extract are certainly synergistic and derived from the whole phytochemical content, as these two phenylpropanoid glycosides make up a significant part of the whole non-volatile phytochemical content, it would be a reasonable to assume that the beneficial effects are largely tied to the beneficial properties of these two phytochemicals. This hypothesis is further discussed in Section 2.4.

### 2.3. Identification and Quantification of Volatile Phytochemicals

All volatile phytochemicals comprising more than 1.0% of the total volatile content detected in the *Plantago* samples are given in Table 4. Most of the compounds quantified in this study were also found in a previously published paper, where the essential oil from *P. lanceolata* was isolated using a hydrodistillation procedure [44]. Our HS-SPME-GC-MS results demonstrate that the volatile phytochemical in excess in all the extracts analysed is β-caryophyllene, with the highest concentration apparent in the sample of *P. major* leaves, PMFM, at 38.9%, and in all other samples, over 20%. Aside from the standout high concentration in PMFM, there is no substantial difference in the β-caryophyllene concentration between other samples. β-caryophyllene is reported to exhibit several biological activities, including antioxidant and anti-inflammatory [45]. D-limonene is present in a sample of *P. major* leaves, PMFK, and two samples of *P. lanceolata* leaves, PLFM and PLFJ, as the volatile compound with the second highest concentration, at 6.0%, 10.3%, and 11.6%, respectively, and as the volatile compound with the third highest concentration in two samples of *P. major* leaves, PMFM and PMFK, at 6.9% and 6.0%, respectively. D-limonene has previously been found to exhibit antioxidant and anti-inflammatory effects [46]. α-caryophyllene was found in the second highest concentration in the three *P. major* samples, PMFM, PMFK, and PMHK, at 9.1%, 6.0%, and 6.1%, respectively, and in the third highest concentration in two *P. lanceolata* samples, PLFJ and PLHS, at 5.5% and 5.6%, respectively. α-caryophyllene is thought to exhibit antibacterial effects [45]. Anethole was found in the second highest concentration in the *P. major* sample PMHK (tied with carvone), and *P. lanceolata* samples, PLFM, and PLHS, at 4.0%, 6.0%, and 7.4%, respectively. Anethole has been demonstrated to exert antimicrobial and antioxidant properties [47]. Therefore, the beneficial therapeutic activities tied to the main volatile compounds present in the *P. major* and *P. lanceolata* samples have strengthened the hypothesis that these plants are well-suited to novel therapeutic approaches.

### 2.4. Evaluation of B. burgdorferi Inhibiting Activity

The residual viability percentages of the stationary-phase *B. burgdorferi* culture after a week of incubation with the *Plantago* extracts are presented in Figure 2. All extracts exhibited *B. burgdorferi* viability reducing activity. The extracts that showed the strongest inhibitory effects against the bacteria were the extract of *P. major* leaves, PMFM, and the extracts of *P. lanceolata* leaves, PLFM and PLFJ, at 17.6 ± 15.8%, 23.6 ± 6.5%, and 14.7 ± 0.5% residual viability, respectively. There was no statistically significant difference noted between the three best *B. burgdorferi* stationary-phase culture inhibitors. When comparing the antibacterial effectiveness results with the phytochemical content of the extracts (Figure 3), it is evident that the three extracts richest in phytochemicals (PMFM, PLFJ, and PLFM) are also the ones most potent against stationary-phase *B. burgdorferi*. Both the extract with the highest total phytochemical content and the highest acteoside content (PLFM) and the extract with the highest plantamajoside content (PMFM) are amongst the most efficient antibacterials, as well as the extract containing the highest number of unique non-volatile phytochemicals (PLFJ). To test the antibacterial potential of plantamajoside and acteoside individually, stationary-phase *B. burgdorferi* were incubated with the ethanolic solutions of the reference standards at concentrations comparable with the highest concentrations these compounds were determined at in the plant material, i.e., 1750 mg/L (comparable to PMFM) and 3500 mg/L (comparable to PLFM), respectively. The results, shown in Figure 2, demonstrate that both compounds showed antibacterial activity (which was concentration-dependent when lowering the respective doses). The inhibitory effect of plantamajoside was not at the level of PMFM. However, there was no statistically significant difference between the antibacterial efficacy of acteoside and PLFM. Considering that the two other extracts that were among the most effective antibacterials did not contain nearly as high concentrations of acteoside, this compound alone could not be found responsible for the *B. burgdorferi* inhibiting properties of the *Plantago* extracts. This would lead to the conclusion that the *B. burgdorferi* inhibiting effects are exerted most beneficially by the synergistic action of a variety of phytochemicals present in the *Plantago* extracts. The most effective extracts also contained the highest numbers of individual non-volatile phytochemicals, PLFJ (23 in total), followed by PMFM and PLFM (both at 13). The three other extracts, PMFK, PMHK, and PLHS, exhibiting the lowest numbers and concentrations of phytochemicals, also exhibit the least amount of antibacterial activity (with no statistically significant differences between these efficacies). Therefore, it is of vital importance that any future therapeutic approaches utilise freshly gathered plant material to maximise the antibacterial potential of *P. major* and *P. lanceolata*.

The *B. burgdorferi* inhibiting effects of the *Plantago* extracts were then evaluated on the biofilm fraction. The biofilm-specific antibacterial effects of the extracts and the two phytochemicals, plantamajoside and acteoside (in concentrations comparable to the respective highest concentrations found in *Plantago* extracts), are shown in Figure 4. The most effective biofilm growth inhibitors were the PMFM and PLFM extracts, both reducing the biofilm volume up to 30%. These two extracts also exhibited the strongest antibacterial effects on the whole stationary-phase culture. The third extract with the strongest antibacterial activity on the whole *B. burgdorferi* culture, PLFJ, did not reduce biofilm formation. It seems that, similarly to the antibacterial properties on the whole stationary-phase culture, the biofilm-specific activity is driven by the compound effect of the phytochemical mixture of the whole crude extract, as plantamajoside and acteoside individually did not succeed in lowering the biofilm content to a similar degree as the extracts with the highest concentrations of these compounds, i.e., PMFM and PLFM.

When comparing the efficacy of the extracts on the whole stationary-phase culture to their biofilm reducing power, it is evident that these two effects are not correlated. Our results are in accordance with the previously demonstrated notion that the agents that reduce biofilm formation do not necessarily affect bacterial viability, as these two inhibitory functions are driven by a variety of different molecular mechanisms [48]. Therefore, it is even more noteworthy that the extracts with the highest phytochemical content amongst both *P. major* and *P. lanceolata* samples, PMFM and PLFM, showed potential as biofilm reducing agents in addition to their efficacy as bacterial viability reducers. Both *P. major* and *P. lanceolata* leaves are thus valuable sources for future research into novel solutions for chronic Lyme disease. When considering the therapeutic potential of *P. major* and *P. lanceolata*, we have demonstrated that these plants contain useful antibacterial compounds, some of which additionally aid in the inhibition of *B. burgdorferi* biofilm formation.

## 3. Materials and Methods

### 3.1. Chemicals and Materials

Ultrapure water (≥18 MΩcm) produced within the laboratory with a Milli-Q water purification system (Merck KGaA, Darmstadt, Germany) was used to prepare all aqueous solutions and as an HPLC eluent. The other HPLC eluent, acetonitrile (≥99.9%), was purchased from Honeywell (Offenbach am Main, Germany). Formic acid (≥99.0%), used in HPLC eluents, was purchased from Fisher Chemical (Pardubice, Czech Republic). Extraction solvent ethanol (96.7%) was obtained from Magnum Veterinaaria AS, Harjumaa, Estonia. Total polyphenolic content of the extracts was evaluated using the 2 M Folin–Ciocalteu reagent, purchased from Sigma-Aldrich (Buchs, Switzerland), and water-free sodium carbonate, purchased from Sigma-Aldrich (Taufkirchen, Germany). Standard solutions used were prepared from gallic acid monohydrate (Sigma-Aldrich, Beijing, China) and 96.7% ethanol (Sigma-Aldrich, Taufkirchen, Germany). For the detection of total flavonoid content, aluminium chloride, obtained from Fluka (Buchs, Switzerland), was used. Standard solutions were prepared from quercetin (≥99.0%, Lachema/Chemapol, Brno, Czech Republic) and methanol (≥99.9%, Honeywell, Charlotte, NC, USA). Total iridoid content was determined using the Trim-Hill reagent, prepared from water-free acetic acid (≥99.0%, Sigma-Aldrich, Taufkirchen, Germany), 37% hydrochloric acid (Honeywell/Fluka, Wien, Austria), and copper sulfate pentahydrate (Sigma-Aldrich, Taufkirchen, Germany). Standard solutions were prepared using aucubin (≥98%) purchased from Cayman Chemical, Ann Arbor, MI, USA. Fluorescein sodium salt (≥98.5%) and AAPH (2,2′-azobis(2-methylpropionamidine)dihydrochloride, 97%), used in the antioxidativity studies, were purchased from Fluka (Buchs, Switzerland) and Sigma-Aldrich (Taufkirchen, Germany), respectively. Standard solutions were prepared from Trolox (6-hydroxy-2,5,7,8-tetramethylchroman-2-carboxylic acid, 97%), purchased from Sigma-Aldrich (Taufkirchen, Germany). Standard compounds used for phytochemical quantification, plantamajoside (99.5%) and acteoside (99.8%), were purchased from MedChemExpress, Monmouth Junction, NJ, USA. Chlorogenic acid (≥95%), luteolin (≥98%), apigenin (≥95%), and the internal standard, bicalutamide (≥99.8%), were purchased from Sigma-Aldrich (Taufkirchen, Germany).

### 3.2. Plant Extracts

Plant samples were named PMFM, PMFK, PMHK, PLFM, PLFJ, and PLHS (abbreviations explained below), chosen to represent three samples of *P. major* and *P. lanceolata* each, with both selections containing two samples of leaves and one of herb (the whole aerial part of the plant). Plant material for samples PMFM, PLFM, and PLFJ was gathered from private gardens in July of 2022. For samples PMFM and PLFM, *P. major* leaves and *P. lanceolata* leaves, respectively, were obtained from Matsalu, Estonia. Sample PLFJ, *P. lanceolata* leaves, was obtained from Jälgimäe, Estonia. The identification for the plant samples of PMFM, PLFM, and PLFJ was performed by Toomas Kukk, and voucher specimens (TAA0166882–TAA0166884) were deposited in the Herbarium of the Institute of Agricultural and Environmental Services (TAA) of the Estonian University of Life Sciences. Plant material for samples PMFK and PMHK was purchased from Kubja Ürditalu, Estonia, as *P. major* leaves and herbs, respectively. Sample PLHS, *P. lanceolata* herb, was purchased from Salus, Germany. The self-gathered plant material was air-dried at room temperature. The bought samples had been previously dried. The moisture content of all samples was measured using a moisture analyser by Ohaus (Parsippany, NJ, USA) and determined as follows: PMFM—7.50%; PMFK—6.89%; PMHK—5.68%; PLFM—6.48%; PLFJ—7.16%; PLHS—6.07%. All plant material studied was kept at an ambient temperature in a dark space before the extraction procedure.

The dried plants were finely ground using a coffee bean grinder Bomann KSW 445 CB (Jinhua, China). Extraction was carried out using 60% ethanol as a solvent; the ratio of plant powder to solvent was 1:20 (*w*/*v*). The extraction procedure consisted of 30 min of shaking using Orbital Shaker DOS-20M (Riga, Latvia) at 250 r/min (with a direction change after 99 r), then sonication at 640 W (350 kHz) in Sonorex^TM^ Digital 10P bath (Bandelin, Berlin, Germany) for 30 min at 35 °C. The extract was then vacuum filtered through a Sartorius^TM^ 3 h filter (70 mm, 65 g/m^3^) (Sartorius, Aubagne, France). The extracts were centrifuged before any experimental use and kept in a dark refrigerator at 4 °C in between experiments.

### 3.3. Colorimetric Analyses

The total concentrations of groups of bioactive compounds in the plants were quantified via colorimetric tests: polyphenols via Folin–Ciocalteu [49], flavonoids via AlCl_3_ [50], and iridoids via the Trim–Hill method [51]. All colorimetric analyses were conducted on the Varian Cary 50 Bio UV-Vis spectrophotometer (Agilent Technologies, Santa Clara, CA, USA), either using 1.5 mL plastic cuvettes (NovaNatura, Casaleggio Novara, Italy) in the analyses of polyphenolics and flavonoids or using 0.7 mL quartz cuvettes (Hellma, Müllheim, Germany) in the analyses of iridoids. The total polyphenolic content of the extracts was determined using calibration solutions of 10, 25, 50, 75, and 100 mg/L of gallic acid in ethanol, prepared from a 5 g/L stock solution. The calibration solutions for the quantification of flavonoid contents were prepared in concentrations of 2, 5, 10, 20, and 40 mg/L of quercetin in methanol, prepared from a 2 g/L stock solution. The calibration solutions for iridoid content measurements were prepared in concentrations of 100, 200, 400, 800, and 1000 mg/L of aucubin in 60% ethanol from a 1 g/L stock solution. All samples were measured in triplicate, and the results were given in mg of either mean gallic acid, quercetin, or aucubin equivalents per g of dried plant material (mg GAE/QE/AE per g) ± standard deviation (Table 2), calculated based on the ratio of plant material (g) to solvent (mL) used for extraction (1:20).

### 3.4. Antioxidative Activity Measurements

The antioxidative activity of all extracts was evaluated using the ORAC_FL_ (oxygen radical absorbance capacity) method with minor modifications as described by Naguib [52]. The 24.25 mM fluorescein stock solution, 30 mM Trolox stock solution, and 600 mM AAPH solution were all prepared daily in a 100 mM phosphate buffer (pH = 7.4). The total volume of the reaction mixture was 3 mL; the mixture was composed of 2.7 mL of 0.5 nM fluorescein (diluted in phosphate buffer) and 100 µL of sample or Trolox dilution, which was incubated at 37 °C for 3 min; then, 200 µL of 600 mM AAPH was added. The samples were measured with a Hitachi F-7000 Fluorescence Spectrophotometer (Chiyoda, Tokyo, Japan) at λex/em 495/520 nm, slits 5 nm, and the time scan was recorded for 3000 s once per s. The calibration solutions were prepared from a diluted 300 µM Trolox solution in final concentrations of 0.5, 1, 2, 4, 6, 8, and 10 µM in the reaction mixture, and the calibration was given as area under curve change from 0 µM Trolox (blank sample). All extracts added to the reaction mixture had previously been diluted 500-fold in the phosphate buffer. Due to the matrix effects apparent in the diluted samples of 60% ethanol extracts, the calibration samples were prepared in 60% ethanol diluted 500-fold in phosphate buffer. All samples were measured in triplicate and the results given in mg of mean Trolox equivalents per g of dried plant material (mg TE/g) ± standard deviation (Table 2), calculated based on the ratio of plant material (g) to solvent (mL) used for extraction (1:20).

### 3.5. HPLC-DAD-MS Analyses

The 1 mg/mL standard solutions of chlorogenic acid, luteolin, apigenin, and 5 mg/mL solutions of plantamajoside and acteoside were prepared in ethanol and diluted to create calibration curves for the quantitative HPLC-DAD analyses. The calibration curves were constructed based on the UV signal at 254 nm (slit 4 nm) using the relative peak area (ratio of the peak area of a standard compound to the peak area of the internal standard bicalutamide). A total of 2 g/L internal standard ethanolic solution was added at a final concentration of 40 mg/L into all the samples. Calibration curves for chlorogenic acid, luteolin, apigenin, and plantamajoside were constructed in the concentration ranges of 5 to 250 mg/L. The calibration curve for acteoside ranged from 10 to 500 mg/L. The phytochemicals on the chromatograms that were not identified using a reference standard were quantitatively measured using one of these five calibration curves by matching the compound with a reference standard based on the similarities in the respective UV absorption spectra (measured in the range of 200 to 400 nm). These selections and respective concentrations are given in Table 3. The chromatographic peaks with a relative peak area (RPA) of less than 0.2 on all chromatograms of the different extracts were excluded from Table 3. All chromatographic constituents that were quantified had an RPA of 0.2 for at least one of the extracts. When another extract showed a non-quantifiable peak (RPA<0.2) with the same retention time (and MS/MS fragmentation pattern), it was marked as “<LOQ” in Table 3; and when there was no peak detected (RPA<0.1), it was marked as “n.d.” in Table 3.

Before the HPLC-DAD-MS/MS analyses, the plant extracts were centrifuged; then, the sample was diluted in ultrapure water two-fold and spiked with the internal standard bicalutamide at a final concentration of 40 mg/L. The sample volume injected was 5 µL. HPLC-DAD-MS/MS analyses were conducted using an Agilent 1260 Infinity II instrument (Agilent Technologies, Santa Clara, CA, USA) with an Agilent Poroshell 120 EC-C18 column—particle size: 2.7 µm; measurements 4.6 × 100 mm (Agilent Technologies, Santa Clara, CA, USA)—thermostated at 28 °C. The mobile phase consisted of ultrapure water (A) and acetonitrile (B), both acidified with 0.1% (*v*/*v*) formic acid. The elution procedure was a linear gradient increasing from 5% to 50% B (0–20 min), then from 50% to 95% B (20–25 min), isocratic 95% B (25–30 min), a linear gradient decreasing from 95% to 5% B (30–30.01 min), and isocratic 5% B (30.01–35 min). The flow rate was kept at 0.6 mL/min. The column was coupled with an Infinity 1260 DAD (Agilent Technologies, Santa Clara, CA, USA), the chromatograms were recorded at the UV-absorbance wavelength of 254 nm (slit 4 nm), and DAD spectra were recorded in the range of 200 to 400 nm. Following the DAD analysis, the sample was analysed using the LC/MSD Trap XCT mass spectrometer (Agilent Technologies, Santa Clara, CA, USA) equipped with an electrospray ionization source. The mass spectra were recorded in negative-ion mode in the m/z range from 100 to 1000. Nitrogen was used as the nebulizing and drying gas, and helium served as the collision gas. The MS/MS fragmentation patterns (generated using the automatic Bruker Daltonics DataAnalysis 5.0 software MS/MS settings) were used to identify, tentatively, the compounds for which there were no reference standards available.

### 3.6. HS-SPME-GC-MS Analyses

The SPME procedure was performed on PDMS/DVB Stabile Flex fiber (poly-dimethylsiloxane/divinylbenzene coating thickness 65 μm, Supelco, Bellefonte, PA, USA) using a manual SPME fiber holder (Supelco-57330-U). SPME fiber was conditioned according to the manufacturer’s instructions prior to the first use. A total of 50 mg of the dried and powdered sample was placed into a 1.5 mL glass vial and closed. The vials were thermostated for 15 min at 50–55 °C to perform headspace extraction of volatiles from plant material. The fiber was then withdrawn from the needle and inserted into the GC injection port, where the analytes were thermally desorbed for the GC-MS analysis. Chromatographic separations were performed on an Agilent Technologies (Santa Clara, CA, USA) 7890A GC system equipped with an ultra-inert splitless liner (Agilent Technologies, type 5190-2293). The gas chromatograph was coupled to an Agilent 5975C mass spectrometer with an electron ionization source and a quadrupole mass analyser. The flow rate of carrier gas (helium 6.0, AGA, Tallinn, Estonia) was kept constant at 1.2 mL/min, and compounds were separated in a ZB-5plus capillary column (30 m × 0.25 mm × 0.25 μm, Agilent Technologies, Santa Clara, CA, USA). The injector temperature was kept at 275 °C, injection was performed in the splitless mode for 2 min. The following oven temperature program was used: the initial temperature was 35 °C; then, it was increased to 200 °C (5 °C min^−1^) and then to 280 °C (20 °C min^−1^, held for 2 min). The total run time was 39 min, starting from fiber introduction into the injection block. The analyte ionization was performed in electron ionization mode using the electron energy of 70 eV. The interface, ion source, and mass analyser temperatures were set at 280, 230, and 150 °C, respectively. Scan mode in the range of 20–500 *m/z* was used for monitoring all analytes. All samples were analysed thrice for confirmation. All compounds were determined by the National Institute of Standards and Technology 17 (NIST 17) library, and Agilent MassHunter Qualitative, Quantitative, and Unknowns Analysis was used for data analysis. Compounds identification relied on achieving an NIST MS library match factor score of ≥800 and a MassHunter Unknowns Analysis software (version 12.0) match factor ≥80. To ensure identification confidence, an experienced analyst visually inspected the mass spectra of all compounds discussed in this paper.

### 3.7. Antibacterial Activity Evaluation

Low passage isolates (≤8) of *B. burgdorferi* strain B31 were obtained from the American Type Culture Collection (Manassas, VA, USA). *B. burgdorferi* was cultured in BSK-H medium with 6.67% rabbit serum. All culture media were filter-sterilised using a 0.2 µm filter. The cultures were incubated in 50 mL sterile closed conical tubes at 33 °C in 5% CO_2_ without antibiotics. After incubation for 7 days, the *B. burgdorferi* culture went into stationary phase (~10^7^ spirochetes/mL) [53], followed by the transferring of the bacterial cultures into 96-well tissue culture microplates for the screening of the *B. burgdorferi* inhibiting activity of the extracts.

To evaluate the activity of *Plantago* extracts against stationary-phase *B. burgdorferi*, the probes were added to 100 µL of the seven-day-old *B. burgdorferi* culture in the 96-well plate; each extract was added to six wells, 5 µL per well. The entire experiment was repeated in triplicate for each sample. All plates were incubated at 33 °C in 5% CO_2_ for the next 7 days. Control cultures (in 6 + 6 wells per plate) were incubated alongside the ones treated with extracts. The effect of antibiotics on the stationary-phase culture was tested, and a solution of a mix of antibiotics doxycycline, daptomycin, and cefoperazone at a final concentration of 50 µM was used as a positive control. The live and dead cells were evaluated using the SYBR Green I/propidium iodide (PI) assay, and the viability percentage was calculated through a generated regression equation. To estimate the residual viability of *B. burgdorferi* after incubation with the extracts, the SYBR Green I/PI assay was performed as described by Feng et al. [53]. Briefly, 5 µL SYBR Green I (100 × stock, Invitrogen, Waltham, MA, USA) and 5 µL propidium iodide (0.5 mM, Sigma, St. Louis, MO, USA) were each added to three wells per extract. The plates were incubated in the dark for 15 min at 33 °C. Fluorescence measurements were taken, with λex at 450 nm, λem at 535 nm (green emission) and 635 nm (red emission) for wells containing SYBR Green I and PI, respectively, of the screening plate using a TECAN Genios Pro microplate reader (Männedorf, Switzerland). Additionally, *B. burgdorferi* suspensions at five different proportions (0:10, 2:8, 5:5, 8:2, 10:0) of live/dead cells were mixed and each added to 6 wells of the 96-well plate. Then, the SYBR Green I and PI reagents were each added to three wells per each of the five samples, and the green/red fluorescence ratios for each sample of live/dead *B. burgdorferi* were measured. Using least-square fitting analysis, the regression equation and regression curve of the relationship between the percentage of live bacteria and the ratios of green/red fluorescence signals were obtained. The regression equation was used to calculate the percentage of live cells in each well of the screening plate, resulting in the residual viability estimation.

The efficacy of the extracts on biofilm was determined by quantifying the biomass fraction, using crystal violet staining as previously published by Woolard et al., with some modifications [54]. Following the incubation period, the media were slowly discarded, leaving behind the attached biofilms on the surface of the plate. Biofilm plates were then submerged in PBS to wash away any remaining non-adherent bacteria and heat fixed (for 1 h at 60 °C). Biofilms were then stained with a 2% crystal violet solution for 5 min and washed twice via submergence into ultrapure water before the bound dye was released with 33% glacial acetic acid. Biofilm growth was then quantified by measuring the optical density of the solubilized dye at 570 nm using the TECAN Genios Pro microplate reader, normalising the biofilm volume in the control sample as 100%.

### 3.8. Statistical Analysis

The statistical analysis of antibacterial efficacy results was performed using one-way analysis of variance (ANOVA) with Bonferroni’s multiple comparison testing. All measurements represent data from at least three independent experiments, given as mean ± standard deviation. The statistical analyses were performed using GraphPad Prism version 10.2.3.

## 4. Conclusions

Three samples of both *P. major* and *P. lanceolata* were studied to determine their phytochemical content, antioxidant potential, and specific antibacterial effect against both stationary-phase *B. burgdorferi* culture and its biofilm fraction. It was demonstrated that these plants contain high amounts of polyphenolics and exhibit considerable antioxidant capacity. There have been several reports on the beneficial properties of both the main non-volatile (plantamajoside, acteoside) and volatile compounds (β-caryophyllene, D-limonene, α-caryophyllene) found in the *Plantago* samples, including antioxidant, anti-inflammatory, and antibacterial effects; thus, these findings indicate the potential of these plants in antibacterial therapeutic approaches. The specific antibacterial testing on *B. burgdorferi* stationary-phase culture revealed that the residual viability of the bacteria was as low as 15%, and that both plantamajoside and acteoside also exhibit *B. burgdorferi* inhibiting properties. Lastly, the biofilm formation inhibiting activity of the extracts was tested. Compared to the control sample, the *B. burgdorferi* biofilm volume was reduced by 30% for two extracts, prepared from *P. major* and *P. lanceolata* leaves. Therefore, it was demonstrated that *P. major* and *P. lanceolata*, both found in several geographic regions affected by Lyme disease, are promising resources for novel antibacterial therapeutic approaches against the causative agent of the illness, *B. burgdorferi*. Moreover, as the reduction of bacterial viability and inhibition of biofilm formation are exerted through different mechanisms, it is noteworthy that extracts from both *P. major* and *P. lanceolata* leaves exhibited both of these effects simultaneously. These initial results are promising, but further research is needed to fully understand the molecular mechanisms responsible for the antibacterial and biofilm inhibiting properties and to start developing practical therapeutic strategies.

## Figures and Tables

**Figure 1 ijms-25-07112-f001:**
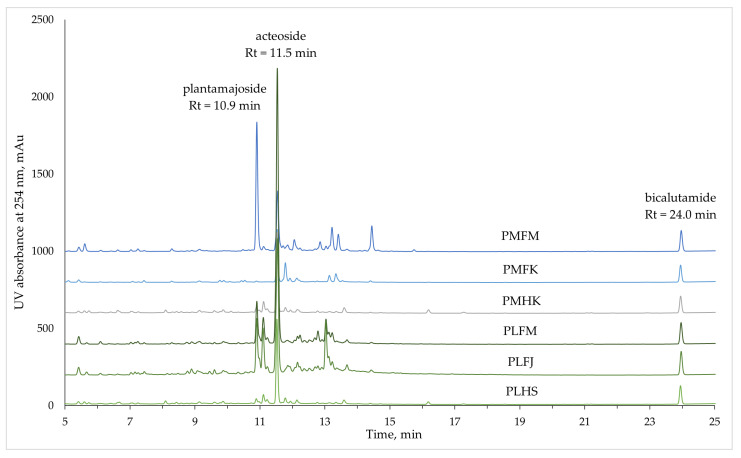
Chromatograms of 60% ethanol *Plantago* extracts twice diluted in water. Plantamajoside and acteoside are the two most significant phytoconstituents of the *Plantago* extracts. Bicalutamide was used as an internal standard at a final concentration of 40 mg/L.

**Figure 2 ijms-25-07112-f002:**
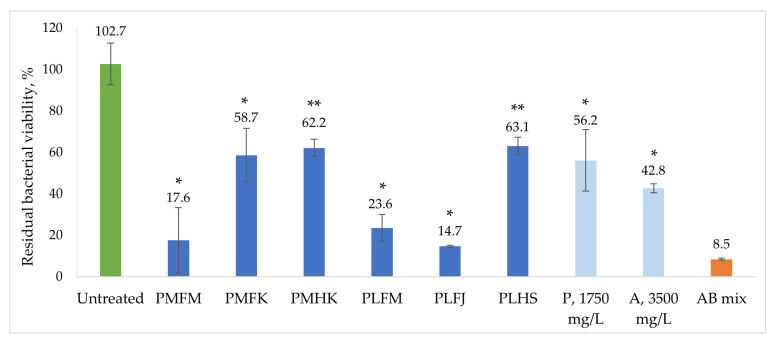
Residual viability percentage of stationary-phase *B. burgdorferi* after a week of incubation with *Plantago* extracts and two phytochemicals: plantamajoside (P) and acteoside (A). Positive control: a mixture of antibiotics doxycycline, daptomycin, and cefoperazone at a final concentration of 50 µM (AB mix). Statistical comparison to the untreated sample: *p*-value < 0.0001 as *, and *p*-value ≤ 0.0002 as **.

**Figure 3 ijms-25-07112-f003:**
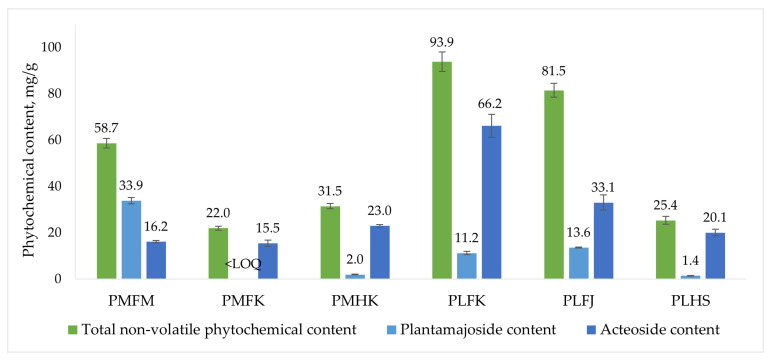
The total quantified non-volatile phytochemical content of the *Plantago* samples, and the concentrations of the two most significant constituents: plantamajoside and acteoside (average of n = 3 experiments along with standard deviation).

**Figure 4 ijms-25-07112-f004:**
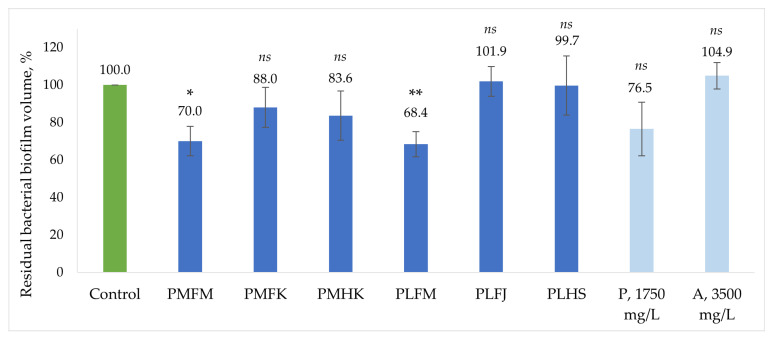
Relative biofilm volume in the *B. burgdorferi* culture after a week of incubation with *Plantago* extracts and two standard compounds: plantamajoside (P) and acteoside (A). Statistical comparison to the control sample: adjusted *p*-value = 0.01 is * and 0.0061 is **; no statistically significant difference is marked as *ns*.

**Table 1 ijms-25-07112-t001:** *Plantago* samples analysed in this study.

Abbreviated Sample Name	Species Name	Part of the Plant Used	Place of Origin
PMFM	*Plantago major* Great plantain	*Plantaginis majoris folium* Great plantain leaves	Matsalu, Estonia
PMFK			Kubja, Estonia
PMHK		*Plantaginis majoris herba* Great plantain herb	Kubja, Estonia
PLFM	*Plantago lanceolata* Ribworth plantain	*Plantaginis lanceolatae folium* Ribworth plantain leaves	Matsalu, Estonia
PLFJ			Jälgimäe, Estonia
PLHS		*Plantaginis lanceolatae herba* Ribworth plantain herb	Salus, Germany

**Table 2 ijms-25-07112-t002:** Antioxidant activity and total polyphenolic, flavonoid, and iridoid content of *Plantago* samples per gram of plant material (average of n = 3 experiments along with standard deviation).

*Plantago* Sample	Antioxidant Activity, mg TE ^1^/g (mmol TE/g)	Total Polyphenolic Content, mg GAE ^2^/g	Total Flavonoid Content, mg QE ^3^/g	Total Iridoid Content, mg AE ^4^/g
PMFM	377.8 ± 11.2 (1.51 ± 0.04)	32.7 ± 2.4	8.3 ± 0.2	5.5 ± 0.9
PMFK	152.7 ± 23.2 (0.61 ± 0.09)	19.7 ± 0.7	5.9 ± 0.6	2.3 ± 0.3
PMHK	300.8 ± 64.5 (1.20 ± 0.26)	25.2 ± 1.7	6.1 ± 0.4	11.4 ± 0.9
PLFM	459.2 ± 78.9 (1.83 ± 0.32)	55.5 ± 4.8	12.4 ± 1.2	23.4 ± 4.6
PLFJ	369.2 ± 85.9 (1.48 ± 0.34)	45.9 ± 0.8	14.0 ± 0.7	10.4 ± 2.1
PLHS	246.3 ± 60.3 (0.98 ± 0.24)	23.4 ± 1.5	4.4 ± 0.3	14.3 ± 1.4

^1^ TE—Trolox equivalents; ^2^ GAE—gallic acid equivalents; ^3^ QE—quercetin equivalents; ^4^ AE—aucubin equivalents.

**Table 3 ijms-25-07112-t003:** UV absorption, mass spectrometric, and concentration data for all quantified non-volatile phytoconstituents in the *Plantago* extracts (average of n = 3 experiments along with standard deviation).

Peak no.	Rt, min	UV Absorption Spectra		Standard Compound for Quantification	Concentration in the Extract, mg/L						Mass Spectra		Tentative Identification
		Max(s), nm	Min(s), nm		PMFM	PMFK	PMHK	PLFM	PLFJ	PLHS	Molecular Ion [M-H]^−^	MS/MS Fragments	
1	5.41	214, 252, 292	234, 276	Apigenin	8.1 ± 0.8	<LOQ ^1^	<LOQ	16.1 ± 0.2	14.7 ± 1.2	<LOQ	807.8	329.0, 477.0, 644.9	
2	5.60	234	212	Luteolin	15.3 ± 1.0	n.d. ^2^	<LOQ	n.d.	n.d.	<LOQ	747.6	373.0	Geniposidic acid dimer
3	8.89	216, 324	262	Apigenin	n.d.	n.d.	n.d.	<LOQ	10.6 ± 0.6	n.d.	878.6	373.1, 521.1, 637.1, 717.2, 799.1	Geniposidic acid derivative
4	9.07	216, 300, 326	262, 304	Apigenin	n.d.	n.d.	n.d.	n.d.	9.9 ± 0.7	n.d.	735.2	544.9, 573.2, 636.1, 699.0, 732.9	
5	10.89	218, 328	262	Plantamajoside	1693.0 ± 65.6	<LOQ	100.3 ± 5.8	562.2 ± 35.1	681.1 ± 12.0	72.4 ± 4.6	639.5	477.3	Plantamajoside ^3^
6	10.97	212, 272, 332	260, 282	Acteoside	n.d.	n.d.	<LOQ	<LOQ	98.3 ± 5.8	n.d.	639.6	477.2	Plantamajoside isomer
7	11.09	248, 268, 348	242, 262, 284	Luteolin	11.8 ± 0.4	n.d.	n.d.	54.4 ± 3.2	93.6 ± 11.2	n.d.	681.5	351.1, 547.0, 663.0	
8	11.11	268, 340	254, 294	Apigenin	n.d.	n.d.	50.4 ± 4.8	n.d.	n.d.	22.2 ± 0.6	637.4	285.8, 461.1	Kaempferol diglucuronide
9	11.22	218, 328	262	Plantamajoside	<LOQ	n.d.	100.0 ± 10.2	88.7 ± 6.3	126.4 ± 11.8	77.4 ± 6.9	755.6	461.9, 593.3, 623.1	Acteoside derivative
10	11.52	218, 330	264	Acteoside	807.9 ± 26.2	773.7 ± 69.0	1150.3 ± 29.8	3309.0 ± 249.8	1652.6 ± 163.5	1004.3 ± 72.0	623.5	461.2	Acteoside ^3^
11	11.71	216, 284, 334	256, 302	Apigenin	10.4 ± 0.5	n.d.	n.d.	n.d.	n.d.	n.d.	639.6	477.2	Plantamajoside isomer
12	11.77	214, 266, 342	262, 282	Luteolin	n.d.	n.d.	25.9 ± 1.9	n.d.	n.d.	14.4 ± 0.9	639.7	315.1, 477.2	Plantamajoside isomer
13	11.78	254, 266, 348	262, 294	Luteolin	n.d.	88.6 ± 7.9	n.d.	n.d.	n.d.	n.d.	923.7	285.0, 461.1	Kaempferol glucuronide
14	11.84	218, 328	262	Plantamajoside	145.2 ± 5.7	n.d.	n.d.	169.8 ± 11.5	n.d.	n.d.	640.0	160.9, 314.9, 477.2, 653.2	Plantamajoside isomer
15	11.85	216, 328	262	Plantamajoside	n.d.	n.d.	n.d.	n.d.	181.4 ± 17.8	n.d.	551.1	533.0	
16	11.91	216, 328	262	Plantamajoside	n.d.	n.d.	n.d.	n.d.	86.6 ± 6.2	n.d.	872.8	625.0, 718.7, 768.9, 827.6	
17	11.92	216, 284, 342	262, 304	Apigenin	n.d.	21.9 ± 2.2	n.d.	n.d.	n.d.	n.d.	565.7	168.9, 322.7, 423.2, 506.0, 528.8, 547.0	
18	12.05	214, 274, 338	262, 298	Apigenin	34.9 ± 1.1	n.d.	n.d.	n.d.	n.d.	n.d.	550.1	531.1	
19	12.08	218, 328	262	Plantamajoside	n.d.	n.d.	n.d.	<LOQ	87.6 ± 8.9	n.d.	756.2	531.5, 593.4, 623.2	
20	12.14	218, 328	262	Plantamajoside	n.d.	137.4 ± 12.9	112.5 ± 9.4	94.7 ± 7.9	144.8 ± 12.4	66.2 ± 5.0	623.5	315.4, 461.2	Acteoside isomer
21	12.22	216, 328	264	Acteoside	<LOQ	n.d.	n.d.	83.5 ± 6.5	61.2 ± 6.8	n.d.	623.5	461.2	Acteoside isomer
22	12.35	218, 332	264	Plantamajoside	<LOQ	n.d.	n.d.	<LOQ	84.1 ± 8.7	n.d.	844.2	513.3, 681.2, 799.3, 825.2	
23	12.51	216, 332	264	Acteoside	<LOQ	n.d.	n.d.	<LOQ	80.5 ± 8.0	n.d.	887.8	557.9	
24	12.69	218, 328	264	Acteoside	<LOQ	n.d.	n.d.	69.0 ± 8.9	83.1 ± 8.4	n.d.	772.1	463.2, 595.1, 639.1, 694.1, 729.3, 753.1	
25	12.78	218, 328	264	Acteoside	n.d.	n.d.	n.d.	n.d.	86.5 ± 14.5	n.d.	736.3	637.1, 687.0	
26	12.79	218, 330	264	Acteoside	n.d.	n.d.	n.d.	146.8 ± 17.5	n.d.	n.d.	637.7	461.3, 491.1	
27	12.85	216, 282, 334	252, 302	Apigenin	26.6 ± 4.8	n.d.	n.d.	n.d.	n.d.	n.d.	563.8	270.8, 336.9, 402.8, 493.1, 519.1, 544.6	Apigenin derivative
28	12.90	218, 326	264	Chlorogenic acid	n.d.	n.d.	n.d.	n.d.	42.2 ± 4.3	n.d.	807.0	623.1, 767.2, 785.3	
29	13.03	246, 270, 336	264, 282	Luteolin	10.6 ± 0.5	n.d.	n.d.	50.2 ± 1.3	98.4 ± 3.1	n.d.	857.7	527.3, 663.1	
30	13.11	248, 270, 334	264, 282	Luteolin	n.d.	n.d.	n.d.	23.3 ± 4.6	34.6 ± 7.8	n.d.	857.7	527.3, 663.1	
31	13.13	218, 266, 334	250, 290	Apigenin	n.d.	35.4 ± 3.0	n.d.	n.d.	n.d.	n.d.	638.3	461.1, 475.1	
32	13.21	212, 274, 334	248, 298	Apigenin	66.0 ± 5.8	n.d.	n.d.	n.d.	n.d.	n.d.	923.9	461.2	
33	13.22	248, 270, 338	264, 280	Luteolin	n.d.	n.d.	<LOQ	26.5 ± 4.9	29.0 ± 5.7	<LOQ	736.1	637.1, 691.0	
34	13.33	214, 250, 266, 344	244, 262, 294	Luteolin	n.d.	43.0 ± 3.6	n.d.	n.d.	n.d.	n.d.	548.1	475.0, 512.8, 621.1	
35	13.36	216, 328	266	Acteoside	n.d.	n.d.	n.d.	<LOQ	77.3 ± 8.5	<LOQ	770.1	593.1, 623.0, 667.1, 764.2	
36	13.41	214, 274, 334	248, 298	Apigenin	41.0 ± 0.9	n.d.	n.d.	n.d.	n.d.	n.d.	547.6	387.0, 426.6, 454.8, 485.0, 530.1	
37	13.58	218, 284, 334	254, 314	Apigenin	n.d.	n.d.	23.3 ± 2.9	n.d.	n.d.	11.6 ± 3.2	531.1	308.8, 353.1, 483.0	
38	13.67	220, 328	272	Plantamajoside	n.d.	n.d.	n.d.	<LOQ	210.3 ± 39.9	n.d.	912.0	411.1, 455.1, 869.4	
39	14.44	214, 274, 332	248, 298	Apigenin	63.0 ± 3.5	n.d.	n.d.	n.d.	n.d.	n.d.	580.0	265.6, 355.0, 518.7	
40	16.18	220, 266, 346	262, 282	Luteolin	n.d.	n.d.	10.0 ± 1.0	n.d.	n.d.	<LOQ	285.3	107.0, 150.8, 174.9, 198.8, 214.7, 242.9	Luteolin
**Total Concentration of Phytochemicals in the Extracts, mg/L**					2934.0 ± 103.4	1100.0 ± 42.7	1572.9 ± 57.9	4694.1 ± 211.3	4074.8 ± 150.3	1268.5 ± 83.1			

^1^ <LOQ—relative peak area under 0.20; the corresponding peak was not quantitatively determined. ^2^ n.d.—relative peak area under 0.10; the corresponding peak was not qualitatively determined. ^3^ Phytochemical identified using a corresponding reference standard.

**Table 4 ijms-25-07112-t004:** Determined volatile compounds in the samples of the *Plantago* species (average of n = 3 experiments along with standard deviation).

Name of the Compound	Rt, min	PMFM, %	PMFK, %	PMHK, %	PLFM, %	PLFJ, %	PLHS, %
Isovaleric acid	7.18	<1.0	<1.0	3.0	1.8	2.3	1.1
α-Methylbutyric acid	7.50	<1.0	<1.0	1.4	<1.0	-	<1.0
Pentanoic acid	8.06	<1.0	<1.0	<1.0	<1.0	1.1	<1.0
Butyrolactone	8.76	1.0	<1.0	2.4	<1.0	<1.0	2.1
Hexanoic acid	10.81	1.6	4.4	2.4	2.1	3.5	5.1
D-Limonene	12.15	**6.9**	**6.0**	2.3	**10.3**	**11.6**	4.3
Fenchone	13.91	-	-	-	<1.0	<1.0	1.1
Linalool	14.17	1.0	2.5	2.5	2.3	2.8	2.4
Isomenthone	15.79	-	1.0	<1.0	1.0	1.3	1.7
Menthol	16.33	-	1.0	1.1	<1.0	<1.0	2.8
α-Terpineol	16.84	<1.0	1.2	1.3	1.0	1.0	1.2
Thymol methyl ether	17.97	1.4	<1.0	1.4	<1.0	-	<1.0
Carvone	18.30	<1.0	4.6	**4.0**	4.0	4.0	4.4
Anethole	19.42	1.2	5.1	**4.0**	**6.0**	5.2	**7.4**
Thymol	19.52	<1.0	1.8	1.0	1.9	1.6	2.3
β-Caryophyllene	22.67	**38.9**	**22.0**	**22.6**	**23.3**	**23.1**	**20.7**
cis-α-Bergamotene	22.77	1.0	-	-	-	-	-
trans-α-Bergamotene	23.30	3.3	2.4	2.2	2.4	2.4	2.2
cis-β-Farnesene	23.71	2.7	1.7	1.6	1.7	1.6	1.6
α-Caryophyllene	23.85	**9.1**	**6.0**	**6.1**	5.9	**5.5**	**5.6**
Aromandendrene	24.33	<1.0	<1.0	1.3	1.4	-	<1.0
β-Ionone	24.51	1.9	1.9	2.1	1.7	2.0	2.2
β-Selinene	24.63	4.2	3.5	3.4	3.5	2.8	2.9
α-Selinene	24.84	2.1	2.3	1.9	2.1	1.3	1.0
β-Bisabolene	25.03	2.3	3.5	3.3	3.1	2.5	3.0
δ-Cadinene	25.43	1.0	1.8	1.9	1.7	1.4	1.8
Dihydroactinidiolide	25.66	1.3	<1.0	2.0	<1.0	1.7	1.9
Caryophyllene oxide	26.33	1.3	1.3	3.2	<1.0	1.1	2.5

The three highest concentrations of volatile phytoconstituents in each sample are marked in bold.

## Data Availability

The raw data supporting the conclusions of this article will be made available by the authors on request.

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
