# Peer review of "Plantago major and Plantago lanceolata Exhibit Antioxidant and Borrelia burgdorferi Inhibiting Activities"

_ijms, 2024, doi:10.3390/ijms25137112_

Round 1

Reviewer 1 Report

Comments and Suggestions for Authors

The manuscript ijms-3041208 entitled “Plantago major and Plantago lanceolata exhibit antioxidant and B. burgdorferi inhibiting activities”, is a nice work about the antibacterial activity against the complex Borrelia burgdorferi. However, to enhance the quality of this work, I suggest the following corrections:

1. In the title: When a is mentioned for the first time, it should be written entirely. Then, replacing "B." for Borrelia would be best. On the other hand, you can abbreviate the second time you mention Plantago.

2. As a suggestion, the authors could add a row to the end of Table 2 with the combined concentration of phytochemical data for each sample.

3. In line 169, the phrase "Our results confirm the earlier reports…" needs references.

4. Section 2.1 needs some discussion (with references) to compare the antioxidative activity found with other plants.

5. In Table 3, add some match index (Match, R-Match, and/or probability %).

6. I suggest that in the residual viability and anti-biofilm experiments, the authors also use as a positive control an antibiotic known to be effective against Borrelia burgdorferi, such as doxycycline, amoxicillin, azithromycin, etc.

7. Figure 3 is not cited in the text.

8. The authors should re-write the conclusion section, avoiding repeating the results data. The conclusion should highlight the more significant findings, including the authors' points of view about the potential use of these species and whether further studies are necessary to complement the current knowledge on the topic.

Reviewer 2 Report

Comments and Suggestions for Authors

thank you for this paper it is well written and i enjoyed reviewing it

Reviewer 3 Report

Comments and Suggestions for Authors

Article

Plantago major and Plantago lanceolata exhibit antioxidant and B. burgdorferi inhibiting activities

A brief summary

The aim of the work seems interesting but the methodology used is questionable. The paper is written in a highly demanding language and requires numerous revisions in order to fully and clearly communicate the authors' intentions.

Broad comments

1. It is fundamental to correctly name and describe the plant material under examination:

a. It is important to note that plants of the ribwort plantain species (Plantago lanceolata) provide the herbal raw material named Ribwort plantain leaf (Plantaginis lanceolatae folium) which has the official status of herbal medicinal product in Europe. (https://www.ema.europa.eu/en/medicines/herbal/plantaginis-lanceolatae-folium). 

However, plants of the broadleaf plantain species (Plantago major) are sometimes used to obtain the herbal raw material named Plantaginis majoris folium (broadleaf plantain leaf). This raw material currently has no official monograph.

b. The herbal raw material is ribwort plantain leaves (Folium Plantaginis lanceolatae) or the whole plant (Herba Plantaginis). What is the definition of the ‘leaves’ and the ‘aerial part’ of these plants in this work? Does the ‘aerial part’ include the flowers? If it was different plant material, it is unlikely to be collated and compared in this way.

c. The names of the samples studied in this work and their descriptions are extremely confusing. I guess the authors themselves get lost in it, since every time they discuss the results they give the entire sample description at the same time along with the abbreviation. I don't think that's what abbreviations are for. Why isn't there a separate table where this is explained in detail. Table 1 contains the results, so it is not suitable as a place to present the test samples. 

d. Why is the phrase ‘self-gathered plant material’ emphasised so many times in the paper? It gives the impression (unintentional?) that the authors attribute some special additional value to this sample. Is it because they collected it themselves and the purchased one is ‘somehow not so good’?

e. On what basis does the statement ‘P. major and P. lanceolata, both ENDEMIC in several geographic regions’ (lines 593-594) appear?

2. The results of the work could have been interesting and valuable if more attention had been paid to the proper selection and preparation of the herbal raw material. At this point, it is possible that the results obtained are coincidental.

3. The work undoubtedly has the potential for novelty, but methodological concerns may nevertheless be decisive.

4 Regarding methodology:

a. Is there confirmation of the identification of the plants under study (voucher specimen)?

b. The preparation of herbal raw material is well researched and described in the lierature. The leaves are recommended to be harvested during the flowering period. They require proper drying (in thin layers in a well-ventilated dryer heated to 35–40 °C), otherwise they blacken due to the formation of aucubin polymers. It is best dried without the petiole, as this is difficult to dry.

c. Why was a double extraction of the plant material carried out? First by shaking and then by sonication (lines 400-403)?

d. The term ‘crude extract’ refers to the extract obtained directly from production and not subjected to additional treatments (purification, enrichment, etc.). In this work, extracts were not treated in this way, so why use the term ‘crude extract’ when there are no, for example, purified extracts?

5. The conclusions contain an unnecessarily extensive repetition of assumptions and methodology.

6. The literature is selective and quite old, not including, for example, the official monographs of the raw material studied or works about production and processing herbal raw materials.

7. Additional comments and suggestions can be found below.

Specific comments

Line 2. Shouldn't the title include the full name: Borrelia burgdorferi sensu lato?

Line 25. I believe that keywords that have already appeared in the title should not be included in the keywords, so instead of ‘Plantago major’ and ‘Plantago lanceolata’ I propose ‘Lyme disease’ and ‘biofilm inhibiting’. The words ‘phytochemicals’ and ‘plant extracts’, on the other hand, are too general.

Table 1 does not include the parameters and results of the statistical analysis and the number of repetitions (n).

Table 2 Table 1 does not include the parameters and results of the statistical analysis and the number of repetitions (n). Names in the ‘Standard compound for quantification’ column are written in lower case and ‘Tentative identification’ in upper case.

Figure 1. The description contains unnecessary information on sample preparation and assessment of the compounds shown on it. However, there is no description of the other determined compounds. Was the analytical wavelength matched to the electron absorption spectra of the analytes? 

Figure 3. Does this graph replicate the data in Table 2? Figure does not include the parameters and results of the statistical analysis and the number of repetitions (n).

Round 2

Reviewer 1 Report

Comments and Suggestions for Authors

The authors addressed all suggestions.

Reviewer 3 Report

Comments and Suggestions for Authors

The authors have made reasonable additions and corrections, and the article has benefited considerably. It seems that in this form it could be published in a reputable scientific journal.

Despite all this, however, there is still a lack of references in the literature to the herbal monograph on Plantago lanceolata L., folium.